# Endophytic Bacteria Isolated from Tea Leaves (*Camellia sinensis* var. *assamica*) Enhanced Plant-Growth-Promoting Activity

Md. Humayun Kabir [1], Kridsada Unban [1], Pratthana Kodchasee [1], Rasiravathanahalli Kaveriyappan Govindarajan [1], Saisamorn Lumyong [2], Nakarin Suwannarach [2], Pairote Wongputtisin [3], Kalidas Shetty [4] and Chartchai Khanongnuch [1,5,*]

1    School of Agro-Industry, Faculty of Agro-Industry, Chiang Mai University, Mueang, Chiang Mai 50100, Thailand
2    Research Center of Microbial Diversity and Sustainable Utilization, Chiang Mai University, Chiang Mai 50200, Thailand
3    Program in Biotechnology, Faculty of Science, Maejo University, Sansai, Chiang Mai 50290, Thailand
4    Global Institute of Food Security and International Agriculture (GIFSIA), Department of Plant Sciences, North Dakota State University, Fargo, ND 58108, USA
5    Research Center for Multidisciplinary Approaches to Miang, Chiang Mai University, Mueang, Chiang Mai 50200, Thailand
*    Correspondence: chartchai.k@cmu.ac.th; Tel.: +66-53-948-261

**Abstract:** Tea (*Camellia sinensis* var. *assamica*) is a traditional and economically important non-alcoholic beverage-producing plant grown in large plantations in the northern region of Thailand and has a diverse community of endophytic bacteria. In this study, a total of 70 bacterial isolates were isolated from healthy asymptomatic samples of tea leaves from five different tea gardens in Chiang Mai, Thailand. Based on 16S rDNA sequence analysis, these bacterial isolates were taxonomically grouped into 11 different genera, namely *Bacillus*, *Curtobacterium*, *Enterobacter Microbacterium*, *Moraxella*, *Neobacillus*, *Priestia*, *Pseudarthrobacter*, *Pseudomonas*, *Sporosarcina*, and *Staphylococcus*. All these isolates were evaluated for their potential to produce indole-3-acetic acid (IAA), siderophores, and cellulolytic enzymes while having phosphate-solubilizing and tannin tolerance capacity. Most isolated bacterial endophytes belonged to the *Bacillus* genus and exhibited multiple plant-growth-promoting abilities. All bacterial endophytes could produce varied concentrations of the indole-related compounds, and the strain *Curtobacterium citreum* P-5.19 had the highest production of IAA at 367.59 μg/mL, followed by *Pseudarthrobacter enclensis* P-3.12 at 266.97 μg/mL. Seventy-eight percent (78%) of the total isolates solubilized inorganic phosphate, while 77%, 65%, and 52% were positive for extracellular proteases, cellulases, and pectinases, respectively. Remarkably, 80% of the isolates were capable of growth on nutrient agar supplemented with 1% (*w/v*) tannic acid. *C. citreum* P-5.19 and *P. enclensis* P-3.12 were selected for evaluation of plant growth promotion, and it was found that both bacterial endophytes enhanced seed germination rate and improved seedling growth parameters such as fresh and/or dry weight, root length, and shoot lengths of sunflower and tomato seeds. The selected bacterial endophytes isolated from tea leaves in this study could be used in bioformulation for plant growth promotion and advancing sustainable agricultural practices contributing to the decreased use of chemical inputs. This is the first report of an endophytic bacterium, *Pseudarthrobacter enclensis*, being isolated from *C. sinensis*.

**Keywords:** endophyte; *Camellia sinensis*; tannin-tolerant; seed germination; IAA

## 1. Introduction

Endophytes are microorganisms that exist ubiquitously within host plant tissues without causing discernible disease symptoms and produce diverse bioactive compounds beneficial to plants [1]. These bacteria are capable of colonizing living tissues of visibly healthy plants with potential to enhance plant growth, provide improved accessibility

to nutrients, and confer resistances to abiotic stresses, potentially resulting in improved yield [2]. The rhizosphere of plants constitutes the major colonization site for bacterial endophytes where areas around the root, including differentiation zones and the intercellular space of the epidermis, are the preferred sites for endophytic bacterial colonization [3]. Nevertheless, some endophytes may spread to other parts of the plant after initial colonization by moving systematically to penetrate vascular tissues [4]. The endophytic absorption of beneficial microorganisms plays an important role in the growth and development of plants with functional traits that contribute to sustainable agricultural practices [5]. Evidence from a large number of studies has shown that many endophytes are capable of synthesizing hormones and solubilizing different kinds of minerals, thus promoting plant growth, development, and yield [6]. In addition, some endophytic bacteria have been reported to exert antifungal activities via production of different hydrolytic enzymes, including proteases, pectinases, and cellulases, which are implicated in the degradation of fungal cell walls. Some of these bacterial genera, including *Bacillus* [7,8], *Pseudomonas* [9], *Serratia* [10], *Enterobacter* [11], and *Xanthomonas* [12], have been shown to support plant growth and nutrient uptake. Such endophytic microbial-based biofertilizers have been targeted as sustainable alternatives to chemical fertilizers because microbial inoculants are considered less harmful and contain different types of microorganisms capable of improving plant growth as well as combating phytopathogens [13]. These endophytic bacterial species are common, less harmful to their host, and therefore can be used to promote plant growth. The use of endophytes in biofertilizers can thus create a new plant-growth-promoting options in field-crop production with value-added benefits.

Tea (*Camellia sinensis*) is the oldest and most consumed non-alcoholic beverage worldwide and is also a repository of a wide variety of endophytic microorganisms [14]. This beverage is targeted for the health-relevant effects of tea-derived phytochemicals, which have been extensively investigated [15,16]. However, the beneficial endophytic bacteria from tea have not been fully investigated, although some studies have revealed that many endophytic bacteria isolated from tea possess multiple plant-growth-promoting traits, including production of ammonia, indole-acetic acid (IAA), siderophore, extracellular enzymes, and phosphate solubilization potential [6,17]. Traditional Thai tea, Miang, has its own inherent benefits and is grown in major provinces in Thailand, including Chiang Mai, Chiang Rai, Mae Hong Son, Lampang, Nan, and Phrae, which constitute the predominant locations for planting and processing of Miang. Chiang Mai is a mountainous area with a tropical climate located in the upper-northern region of Thailand and is the largest area for Miang plantations in a typical agroforestry system as described by Khanongnuch et al. [16]. The harvested tea leaves used as raw material for Miang producing process were analyzed, and it was found that the tea leaves collected from Pa Pae sub-district, Mae Taeng district, Chiang Mai, showed the highest quality in terms of nutritional components and bioactive compounds among 22 Miang processing areas in North Thailand [18]. Interestingly, traditional Miang plantations do not use chemical pesticides or fertilizers and have evolved naturally among the variety of other forest tree species, which is different from the cultivation of tea in China and other Asian countries [16]. This type of mixed plantation naturally allows or promotes plant and microbial interactions and thus suggests that endophytes can adapt themselves to this unique environment and are more likely to evolve and produce beneficial secondary metabolites [19–21]. So far, no studies have been reported on the microbial diversity and/or functional trait analysis of endophytic bacteria isolated from *C. sinensis* tea plants found in Miang plantation areas. Therefore, this study aimed to explore endophytic bacteria associated with tea plants in the selected tea garden, Pa Pae sub-district, which may provide useful information to explain why such tea plantations achieve high-quality tea leaves without the need for chemical pesticides or fertilizers.

This study therefore describes the isolation of culturable endophytic bacteria from Miang tea leaves sampled from five different Miang plantations. The plant-growth-promoting activities of these isolates, such as production of IAA, siderophores, and phosphate solubi-

lization potential, were investigated. The plant-growth-promoting effects of the selected endophytic bacteria on seed germination were also evaluated.

## 2. Materials and Methods

### 2.1. Sample Collection

Samples of tea leaves were collected from five different tea gardens located at Pa Pae sub-district, Mae Taeng District, Chiang Mai, in Upper North Thailand. The area of the tea gardens was in a mountainous region with elevation ranging from 800 to 1300 m above sea level (19°07′00.3′′ N 98°43′35.0′′ E), with temperature ranging from 12 to 35 °C, and 1134 mm of annual rainfall. These tea plants are cultivated after planting following deforesting. The healthy and mature tea plants were randomly selected, and young leaf samples were collected without any marks or injuries. These leaf samples were used as the source for isolation of endophytic bacteria. All samples were placed in sterile zip-locked bags, carried back to the laboratory in an ice box, and subjected to further processing within 24 h.

### 2.2. Isolation of Endophytic Bacteria

Surface decontamination of the leaf sample was performed according to the method described by Fisher et al. [22] with some minor modifications. Leaf samples (10 leaves from each garden) were washed with running tap water and then distilled water in a laminar airflow hood and were sequentially immersed in 70% ($v/v$) ethanol for 30 s and then 5% ($v/v$) sodium hypochlorite for 3 min. Finally, they were washed four times with autoclaved sterile distilled water. For validation of surface sterilization, the final washed solution was spread and cultured on nutrient agar (NA) medium (HiMedia, Mumbai, India) at 30 °C for 4–5 days. Surface-sterilized leaf samples were fragmented into a size of approximately 1 cm$^2$ using a sterile scalpel. To isolate endophytic bacteria from the sample, the leaf explants (2 explants per leaf or 20 leaf explants for each garden in total) were aseptically transferred and placed on nutrient agar (NA) supplemented with 10 μg/mL nystatin for the fungal growth inhibition. The NA-cultured plates were incubated for 2–5 days at 30 °C to allow endophytic bacteria to grow from the leaf explant [23]. Well-grown bacterial colonies which appeared from the leaf explant were randomly collected for further investigation. Pure culture of each bacterial isolate was separately cultivated on nutrient broth (NB) (HiMedia, Mumbai, India) at 30 °C and 150 rpm for 24 h. Liquid stock culture of each isolate in 50% ($w/v$) glycerol was prepared and kept at −80 °C for the further studies.

### 2.3. Molecular Identification and Phylogenetic Analysis

The isolated bacteria were subjected to 16S rRNA gene sequence analysis for species identification. Endophytic bacterial isolates were cultured in nutrient broth for 72 h at 30 °C, and the genomic DNA was extracted according to method described by Russell et al. [24]. Amplification of the 16S rRNA genes of each isolate was performed by polymerase chain reaction (PCR) using two universal primers 27F (5′-AGAGTTTGATCCTGGCTCAG-3′) and 1525R (5′-AAGGAGGTGWTCCARCC-3′). PCR was carried out using a MG96G thermal cycler (LongGene Scientific Instruments Co., Ltd., Hangzhou, China) under the following conditions: 95 °C hold for 2 min, 35 cycles of 94 and 52 °C for 20 s each, and 1 min 30 s at 72 °C, followed by a final extension for 5 min at 72 °C. The PCR products was analyzed by gel electrophoresis and visualized on 1% ($w/v$) agarose gel electrophoresis in 1× TAE buffer. The amplified PCR products of 16S RNA genes were sent to 1st BASE DNA Sequencing Services (Malaysia) for sequencing. The 16S rDNA sequences were analyzed and determined using the BLAST algorithm of gene bank (http://www.ncbi.nml.nih.gov/blast, accessed on 5 January 2022) and EzBioCloud (http://www.ezbiocloud.net, accessed on 5 January 2022). Multiple sequence alignment was performed using BioEdit 7.0, and the phylogenetic tree was created based on the neighbor-joining method by MEGA version 4.0 software.

### 2.4. Screening of Tea Leaf Endophytic Bacteria for Plant-Growth-Promoting Activities

2.4.1. Indole-Related Compounds Production

Quantitative estimation of indole-3-acetic acid production was made for the endophytic bacteria from tea leaves according to the method described by Bric et al. [25]. The bacterial isolates were grown in nutrient broth supplemented with 1 mg/mL L-tryptophan and incubated at 30 °C at 150 rpm for 72 h. The culture was centrifuged at $10,000 \times g$ for 6 min, and the clear culture supernatant was estimated for indole-related compounds by mixing 1 mL of culture supernatant with 2 mL of Salkowski's reagent (0.5 M $FeCl_3$ solution in 35% perchloric) followed by incubation at room temperature for 25 min. The pinkish color formed was measured for the absorbance at 530 nm wavelength by UV spectrophotometry. Kinetics of total indole compounds production was calculated from a reference standard curve and referred to as indole equivalents per milliliter (indole mg/mL).

2.4.2. Investigation of Siderophore Production

Siderophore production ability of isolated endophytic bacterial isolates was accomplished using chrome azurol sulphonate (CAS) agar medium according to the method of Schwyn et al. [26]. Nutrient agar was gently spread with CAS solution to achieve the final concentration of 60 mL/L. The bacterial inoculum grown at 30 °C for 48 h was spotted on a CAS agar plate and incubated at 30 °C for 72 h, and the positive isolates were assessed by the color change of the media around the colonies from blue to orange.

2.4.3. Phosphate Solubilization Test

Insoluble mineral phosphate solubilization ability of the endophytic bacterial isolates was investigated using the spot inoculation method on Pikovskaya's agar medium [27]. Inoculated plates were incubated for 3 days at 30 °C, and the presence of a clear zone surrounding the colony indicated the phosphate-solubilizing ability. The halo formation was estimated from the length of the total diameter subtracted by the length of colony diameter.

### 2.5. Extracellular Enzyme Production Test

2.5.1. Protease Activity

The protease production capacity of the endophytic bacterial isolates was determined using skim milk agar (SMA) containing 5 g/L pancreatic digest of casein, 1 g/L glucose, 2.5 g/L yeast extract, 28 g/L skim milk, and 15 g/L agar. The single colonies of each bacterial isolates were spotted on SMA plates and incubated at 30 °C for 72 h. The presence of a clear zone formed by skim milk hydrolysis around the colony indicates positive proteolytic activity [28].

2.5.2. Cellulase Activity

To determine the ability of extracellular cellulase production, the endophytic bacterial isolates were tested by inoculating on medium containing 20 g/L $K_2HPO_4$, 5 g/L $NaNO_3$, 2 g/L $MgSO_4.7H_2O$, 1 g/L KCl, 10 g/L NaCl, 5 g/L yeast extract, 5 g/L carboxymethyl cellulose (CMC) powder (Sigma-Aldrich, Steinem, Germany), and 15 g/L agar, pH 7.0. After 7 days of incubation at 30 °C, the cultured plates were stained with 0.1% (*w/v*) Congo red followed by overlaying with 1 M NaCl. The bacterial endophytes which showed the formation of translucent zone around their colonies were denoted as the extracellular cellulase-producing isolates [29].

### 2.6. Tannin Tolerance Characteristics

The tannin tolerance of endophytic bacterial isolates was screened on NA medium supplemented with tannic acid at different concentrations. The pH of the tannic acid solution was adjusted to 7.0, separately autoclaved, and aseptically poured into the melted NA medium to achieve final concentrations of 10, 30, and 50 g/L tannin. A single colony of each bacterial isolate was spotted on to the plates using a sterile toothpick, and the growth of bacterial isolates was observed after 7 days incubation at 30 °C [30].

### 2.7. IAA Quantification by HPLC

Seven isolates which produced high quantities of indole-related compounds were selected based on the results of the estimation of indole-related compounds described previously. High-performance liquid chromatography (HPLC) was used to quantify IAA and related indole compounds synthesized by microbial cultures as described by Kumla et al. [31]. HPLC analysis was carried out on a Shimadzu Prominence UFLC system, coupled with a LC-20 AD pump, a SIL-20ACHT for auto-sampling, a CTO-20 column oven, a CBM-20A system controller, and a SPD-20A photodiode array detector (Shimadzu Co., Kyoto, Japan). A Mightysil RP-18 (250 × 4.6 mm, 5 μm) column was used to separate the sample at 40 °C. The mobile phase involved a solution of 2.5% acetic acid in deionized water, pH 3.8 (adjusted by 10 M KOH) (A) and 80% acetonitrile in deionized water (B). The following gradient program was used: 0–25 min, 0–20% B; 25–31 min, increased to 50% B; 31–33 min, increased to 100% B. The flow rate was 0.5 mL/min, and the detection was performed with absorption at 280 and 350 nm. By comparing the retention time and absorption spectrum to IAA standard, the presence of IAA was determined. The microbial IAA was measured using a calibration curve constructed with each standard.

### 2.8. Plant-Growth-Promoting (PGP) Experiment

#### 2.8.1. Preparation of Bacterial Inoculum for Seed Treatment

Two endophytic bacterial isolates, *C. citreum* P-5.19 and *P. enclensis* P-3.12, were selected for plant-growth-promoting experiments based on the highest IAA production determined by HPLC. The selected bacterial strains were cultured in a conical flask containing 200 mL of yeast peptone broth (HiMedia, Mumbai, India) incubated for 48 h at 30 °C. Bacterial cells were separated by centrifugation at $10,000 \times g$ for 4 min and washed three times with sterile 60 mM phosphate buffer pH 6.8 by centrifugation at $6000 \times g$ for 10 min. Then, cell pellets were resuspended in phosphate buffer to achieve the cell concentration of $10^8$ CFU/mL for use as a seed inoculation [32].

#### 2.8.2. Effects of Selected Bacterial Strains on Seed Germination

For the determination of germination percentage and seedling growth experiments, sunflower and tomato seeds were used and were surface sterilized by dipping in 5% (*w/v*) sodium hypochlorite for 2 min and then washed three times with sterile distilled water. Then, the seeds were soaked in selected bacterial suspension ($10^8$ CFU/mL) and inoculated in a Petri dish (9 cm) on a layer of sterile tissue paper at 25 °C. The experiments were repeated in three replications. Seeds soaked with 60 mM phosphate buffer (pH 6.8) instead of bacterial suspension were used as a control. To maintain sufficient moisture, water was added to the Petri dishes. The percentage of germination was recorded every 24 h. Root and shoot length were measured after the sixth day. The percentage of germination was applied in the calculation of the vigor index according to the following formula: vigor index = [mean shoot length (cm) + mean root length] × germination %.

#### 2.8.3. Statistical Analysis

The experiments were carried out in triplicate (n = 3), and results were recorded as mean ± SD. The data were analyzed using SPSS statistical software package, version 20.0 (Windows), IBM Corporation, New York, NY, USA. Tukey's HSD multiple range post hoc test was used, and values were considered significantly different at $p < 0.05$. Unsupervised machine learning was used to classify the plant growth promotion (PGP) data, and the results were presented as the principal component (PC) plots. The input data and the Python code were provided as supporting information (Python 3.6.10, operated via Jupiter notebook text editor in Anaconda).

## 3. Results

### 3.1. Isolation and Phylogenetic Analysis

The current investigation revealed that culturable endophytic bacteria are abundant in tea plants from Miang plantations. A total of 70 bacterial strains were isolated on the basis of colony morphology from the leaves of tea plants in the regions of Northern Thailand. The results obtained from 16S rRNA sequencing of all 70 bacterial isolates showed similarities when compared with those of reference sequences in GenBank, and the data are presented in Table 1. The 16S rRNA gene sequences of the endophytic bacterial isolates obtained from tea leaves were analyzed, and the phylogenetic tree constructed is shown in Figure 1. The isolated bacterial strains represented 11 different genera: *Bacillus* (50.0%), *Microbacterium* (17.1%), *Pseudarthrobacter* (7.1%), *Staphylococcus* (7.1%), *Priestia* (7.1%), *Cartobacterium* (2.9%), and *Moraxella* (2.9%). However, *Pseudomonas*, *Enterobacter*, *Sporosarcina*, and *Neobacillus* constituted a cumulative 1.4% of the bacterial diversity. Most of the isolated endophytic bacteria (60.0%) belonged to the taxonomic class of Bacilli, represented by the genera *Bacillus*, *Neobacillus*, *Priestia,* and *Sporosarcina*. Actinobacteria belonged to the second prevalent class (27.1%) with the genera *Curtobacterium*, *Microbacterium*, and *Pseudarthrobacter*. In addition, other four strains (12.9%) were classified as γ-proteobacteria.

**Table 1.** The BLAST results, including the closest species of type strain, similarity percentage, full length of 16S rRNA gene sequence, and accession number.

| Isolate | Closest Species | Similarity (%) | Length (bp) | Accession Number |
|---|---|---|---|---|
| P-1.2 | *Pseudomonas stutzeri* ATCC17588 [T] | 99.72 | 1447 | OQ295913 |
| P-1.5 | *Microbacterium testaceum* NBRC 12675 [T] | 99.79 | 1445 | OQ295914 |
| P-1.6 | *Bacillus altitudinis* 41KF2b [T] | 99.86 | 1458 | OQ295915 |
| P-1.7 | *Bacillus altitudinis* 41KF2b [T] | 100.00 | 1449 | OQ295916 |
| P-1.9 | *Bacillus altitudinis* 41KF2b [T] | 100.00 | 1449 | OQ295917 |
| P-1.10 | *Bacillus altitudinis* 41KF2b [T] | 100.00 | 1462 | OQ295918 |
| P-1.11 | *Staphylococcus argenteus* MSHR1132 [T] | 99.93 | 1466 | OQ295919 |
| P-1.13 | *Microbacterium testaceum* BJML01000022 [T] | 100.00 | 1375 | OQ295920 |
| P-1.14 | *Microbacterium testaceum* BJML01000022 [T] | 99.86 | 1428 | OQ295921 |
| P-1.16 | *Bacillus altitudinis* 41KF2b [T] | 100.00 | 1453 | OQ295922 |
| P-1.17 | *Bacillus altitudinis* 41KF2b [T] | 100.00 | 1444 | OQ295923 |
| P-1.18 | *Moraxella osloensis* CCUG 350 [T] | 99.23 | 1423 | OQ295924 |
| P-2.1 | *Bacillus altitudinis* 41KF2b [T] | 100.00 | 1444 | OQ295925 |
| P-2.3 | *Staphylococcus argenteus* MSHR1132 [T] | 99.93 | 1452 | OQ295926 |
| P-2.4 | *Bacillus altitudinis* 41KF2b [T] | 100.00 | 1450 | OQ295927 |
| P-2.5 | *Bacillus altitudinis* 41KF2b [T] | 100.00 | 1450 | OQ295928 |
| P-2.6 | *Bacillus altitudinis* 41KF2b [T] | 100.00 | 1451 | OQ295929 |
| P-2.7 | *Bacillus altitudinis* 41KF2b [T] | 100.00 | 1448 | OQ295930 |
| P-2.8 | *Sporosarcina luteola* Y1 [T] | 98.97 | 1454 | OQ295931 |
| P-2.9 | *Bacillus altitudinis* 41KF2b [T] | 100.00 | 1450 | OQ295932 |
| P-2.10 | *Bacillus altitudinis* 41KF2b [T] | 100.00 | 1445 | OQ295933 |
| P-2.12 | *Bacillus altitudinis* 41KF2b [T] | 100.00 | 1448 | OQ295934 |
| P-2.14 | *Bacillus altitudinis* 41KF2b [T] | 100.00 | 1448 | OQ295935 |
| P-2.16 | *Moraxella osloensis* CCUG 350 [T] | 99.08 | 1410 | OQ295936 |
| P-2.17 | *Bacillus altitudinis* 41KF2b [T] | 100.00 | 1453 | OQ295937 |
| P-2.18 | *Bacillus altitudinis* 41KF2b [T] | 100.00 | 1449 | OQ295938 |
| P-2.19 | *Bacillus altitudinis* 41KF2b [T] | 100.00 | 1449 | OQ295939 |
| P-2.20 | *Bacillus altitudinis* 41KF2b [T] | 100.00 | 1449 | OQ295940 |
| P-2.21 | *Bacillus altitudinis* 41KF2b [T] | 100.00 | 1451 | OQ295941 |
| P-2.22 | *Bacillus mediterraneensis* Marseille-P2366 [T] | 98.31 | 1428 | OQ295942 |
| P-3.1 | *Priestia megaterium* NBRC 15308 [T] | 100.00 | 1450 | OQ295943 |
| P-3.3 | *Bacillus altitudinis* 41KF2b [T] | 100.00 | 1446 | OQ295944 |
| P-3.5 | *Bacillus thuringiensis* ATCC10792 [T] | 100.00 | 1457 | OQ295945 |
| P-3.7 | *Pseudarthrobacter enclensis* NIO-1008 [T] | 99.93 | 1383 | OQ295946 |
| P-3.8 | *Bacillus altitudinis* 41KF2b [T] | 100.00 | 1452 | OQ295947 |

**Table 1.** *Cont.*

| Isolate | Closest Species | Similarity (%) | Length (bp) | Accession Number |
|---|---|---|---|---|
| P-3.9 | *Microbacterium testaceum* NBRC 12675 [T] | 99.72 | 1413 | OQ295948 |
| P-3.10 | *Bacillus thuringiensis* ATCC10792 [T] | 100.00 | 1460 | OQ295949 |
| P-3.11 | *Pseudarthrobacter enclensis* NIO-1008 [T] | 99.93 | 1397 | OQ295950 |
| P-3.12 | *Pseudarthrobacter enclensis* NIO-1008 [T] | 99.93 | 1395 | OQ295951 |
| P-3.13 | *Pseudarthrobacter enclensis* NIO-1008 [T] | 99.78 | 1410 | OQ295952 |
| P-3.14 | *Priestia megaterium* NBRC 15308 [T] | 100.00 | 1451 | OQ295953 |
| P-3.16 | *Bacillus altitudinis* 41KF2b [T] | 99.59 | 1450 | OQ295954 |
| P-3.17 | *Bacillus altitudinis* 41KF2b [T] | 100.00 | 1446 | OQ295955 |
| P-3.18 | *Priestia megaterium* NBRC 15308 [T] | 100.00 | 1445 | OQ295956 |
| P-3.20 | *Bacillus altitudinis* 41KF2b [T] | 100.00 | 1450 | OQ295957 |
| P-3.23 | *Priestia megaterium* NBRC 15308 [T] | 100.00 | 1447 | OQ295958 |
| P-4.2 | *Bacillus altitudinis* 41KF2b [T] | 100.00 | 1451 | OQ295959 |
| P-4.3 | *Microbacterium testaceum* NBRC 12675 [T] | 99.86 | 1430 | OQ295960 |
| P-4.4 | *Bacillus altitudinis* 41KF2b [T] | 99.93 | 1448 | OQ295961 |
| P-4.6 | *Bacillus altitudinis* 41KF2b [T] | 99.93 | 1445 | OQ295962 |
| P-4.7 | *Bacillus altitudinis* 41KF2b [T] | 99.93 | 1445 | OQ295963 |
| P-4.10 | *Curtobacterium citreum* DSM 20528 [T] | 99.93 | 1418 | OQ295964 |
| P-4.18 | *Enterobacter wuhouensis* WCHEW120002 [T] | 99.30 | 1432 | OQ295965 |
| P-4.20 | *Pseudarthrobacter enclensis* NIO-1008 [T] | 99.93 | 1396 | OQ295966 |
| P-5.2 | *Staphylococcus haemolyticus* MTCC 3383 [T] | 100.00 | 1442 | OQ295967 |
| P-5.4 | *Bacillus safensis* subsp. *safensis* FO-36b [T] | 99.79 | 1441 | OQ295968 |
| P-5.5 | *Bacillus altitudinis* 41KF2b [T] | 99.93 | 1446 | OQ295969 |
| P-5.6 | *Staphylococcus argenteus* MSHR1132 [T] | 99.86 | 1452 | OQ295970 |
| P-5.9 | *Staphylococcus argenteus* MSHR1132 [T] | 99.93 | 1453 | OQ295971 |
| P-5.10 | *Microbacterium testaceum* NBRC 12675 [T] | 100.00 | 1427 | OQ295972 |
| P-5.11 | *Neobacillus niacini* IFO 15566 [T] | 99.03 | 1441 | OQ295973 |
| P-5.12 | *Microbacterium testaceum* NBRC 12675 [T] | 99.72 | 1412 | OQ295974 |
| P-5.13 | *Priestia aryabhattai* B8W22 [T] | 100.00 | 1443 | OQ295975 |
| P-5.15 | *Bacillus altitudinis* 41KF2b [T] | 100.00 | 1450 | OQ295976 |
| P-5.17 | *Microbacterium testaceum* NBRC 12675 [T] | 100.00 | 1426 | OQ295977 |
| P-5.18 | *Microbacterium testaceum* NBRC 12675 [T] | 99.72 | 1415 | OQ295978 |
| P-5.19 | *Curtobacterium citreum* DSM 20528 [T] | 99.93 | 1420 | OQ295979 |
| P-5.20 | *Microbacterium testaceum* NBRC 12675 [T] | 99.86 | 1420 | OQ295980 |
| P-5.21 | *Microbacterium testaceum* NBRC 12675 [T] | 99.86 | 1425 | OQ295981 |
| P-5.23 | *Microbacterium arborescens* DSM 20754 [T] | 99.50 | 1416 | OQ295982 |

Note: The superscript letter "T" indicates the type strain of bacteria.

With regard to the number of bacterial endophytes recovered from samples of tea leaves collected from different tea garden in Pa Pae sub-district, Mae Taeng district, Chiang Mai province, Thailand (Figure 2A), the highest number of 18 isolates was found from the P-2 tea garden, while the P-4 tea garden was the lowest at only eight isolates, and the average number was 14 isolates per garden. Among all the endophytic bacterial isolates, representatives of *Bacillus* sp., especially *Bacillus altitudinis*, were obtained in all the tea gardens (Figure 2B), implying that it is a common and prevalent class of endophytic bacterial community in Miang tea leaves. *Microbacterium* sp. and *Staphylococcus* sp. were also isolated in four tea gardens (P-1, P-3, P-4, and P-5 and P-1, P-2, P-4, and P-5, respectively), while *Pseudarthrobacter* sp. was found in the P-3 and P-4 tea gardens, and *Moraxella* sp. was found in the P-1 and P-2 tea gardens (Figure 2A,B).

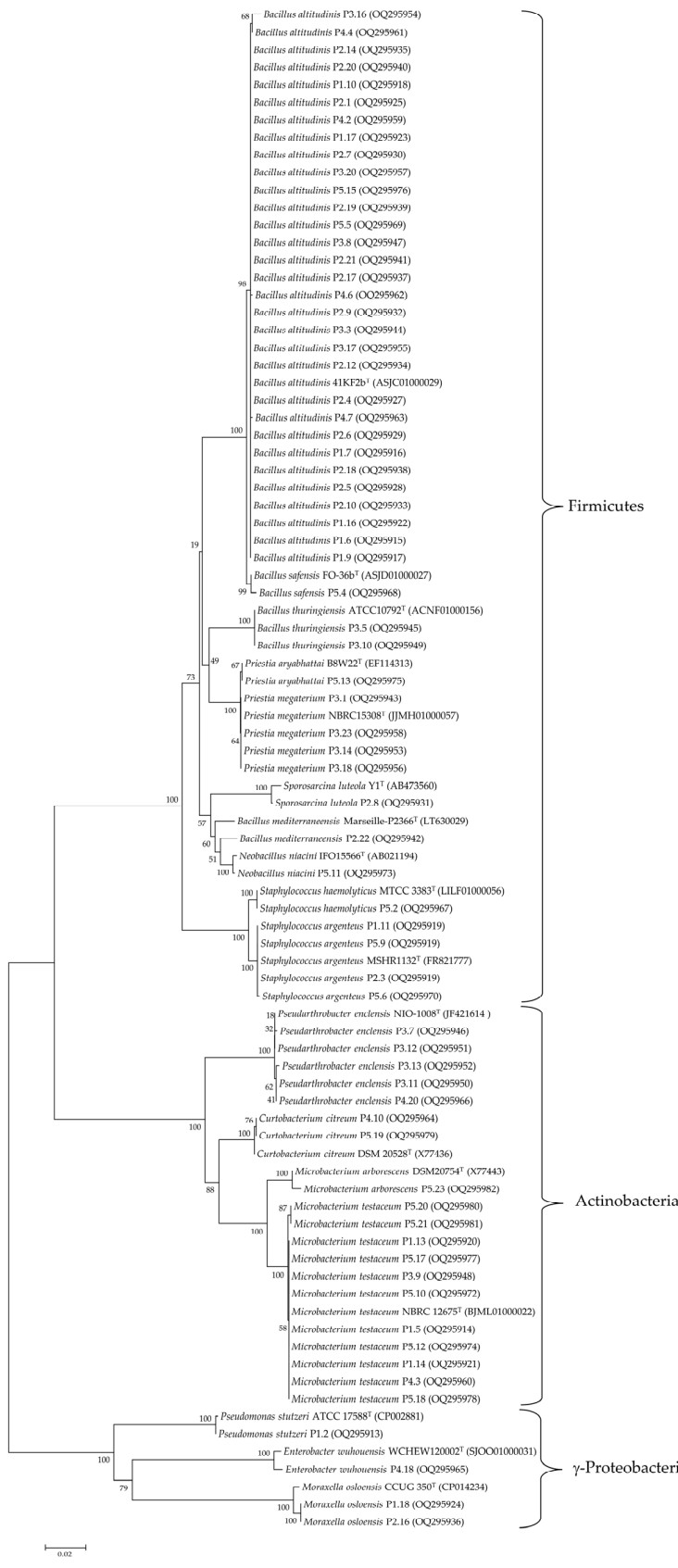

**Figure 1.** Phylogenetic tree based on 16S rDNA sequences of bacterial isolates isolated from tea leaves along with the sequences from reference strains using the neighbor-joining method. The number in each branch shows the bootstrap percentage. Bars represent 0.02 substitutions per nucleotide position. The superscript letter "T" indicates the type strain.

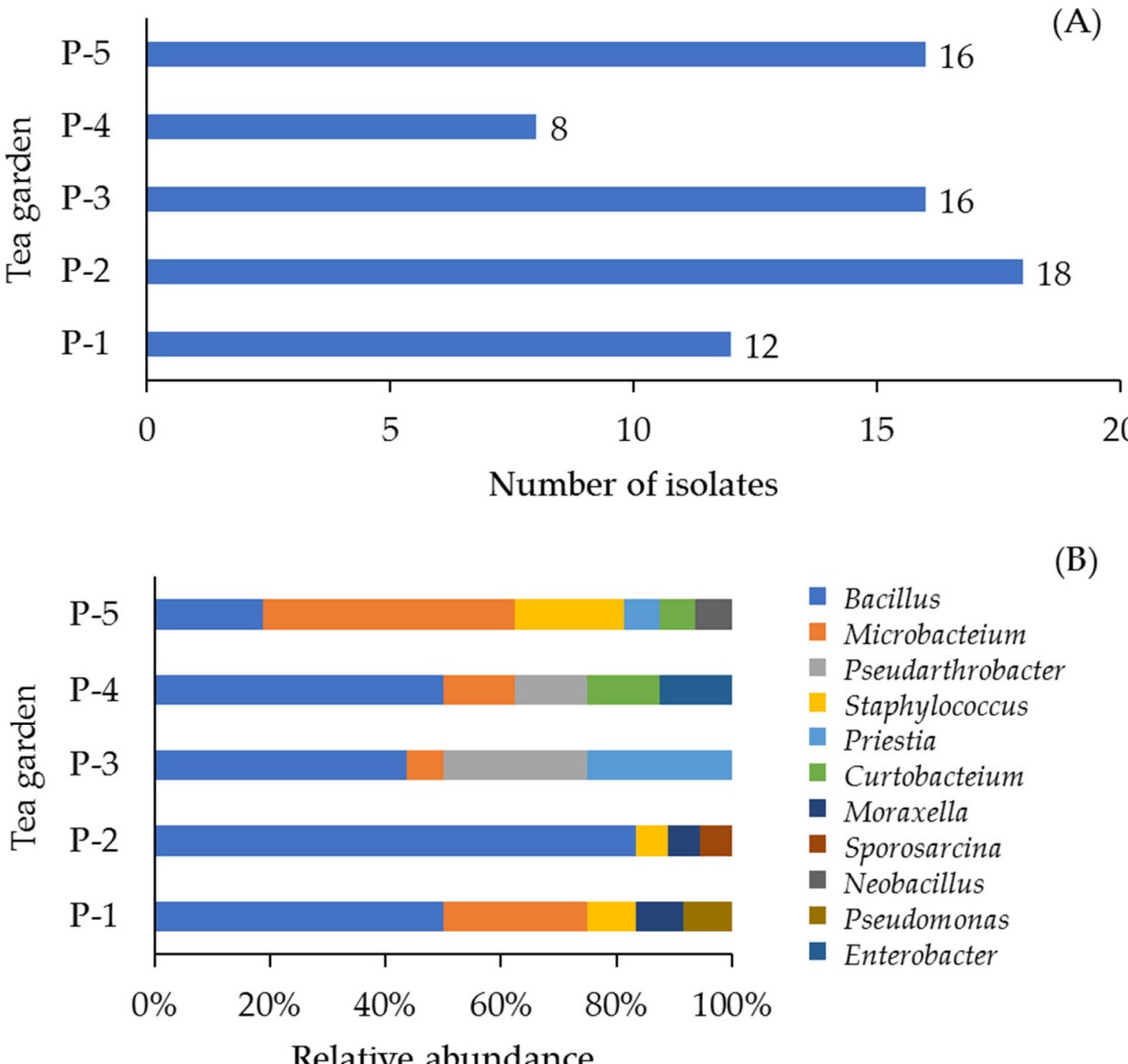

**Figure 2.** Number of endophytic bacterial isolates (**A**) and percent relative abundance of endophytic bacterial genera (**B**) found in each of five different tea gardens of Chiang Mai province.

*3.2. Indole Compounds Production and Accumulation Test*

In the investigation of PGP traits, all endophytic bacterial isolates were able to produce varying concentrations of indole-related compounds (indoles) (4.7–141.5 μg/mL) when growing in the medium containing L-tryptophan. The highest production of indoles was observed in *Pseudarthrobacter enclensis* P-3.12, *Curtobacterium citreum* P-4.10, *Enterobacter wohouensis* P-4.18, and *Microbacterium testacum* P-5.17, which produced 141.5, 133.0, 124.6, and 119.5 μg/mL, respectively (Figure 3). Among all 70 bacterial strains, nine isolates produced notable quantities of indoles (>100 μg/mL).

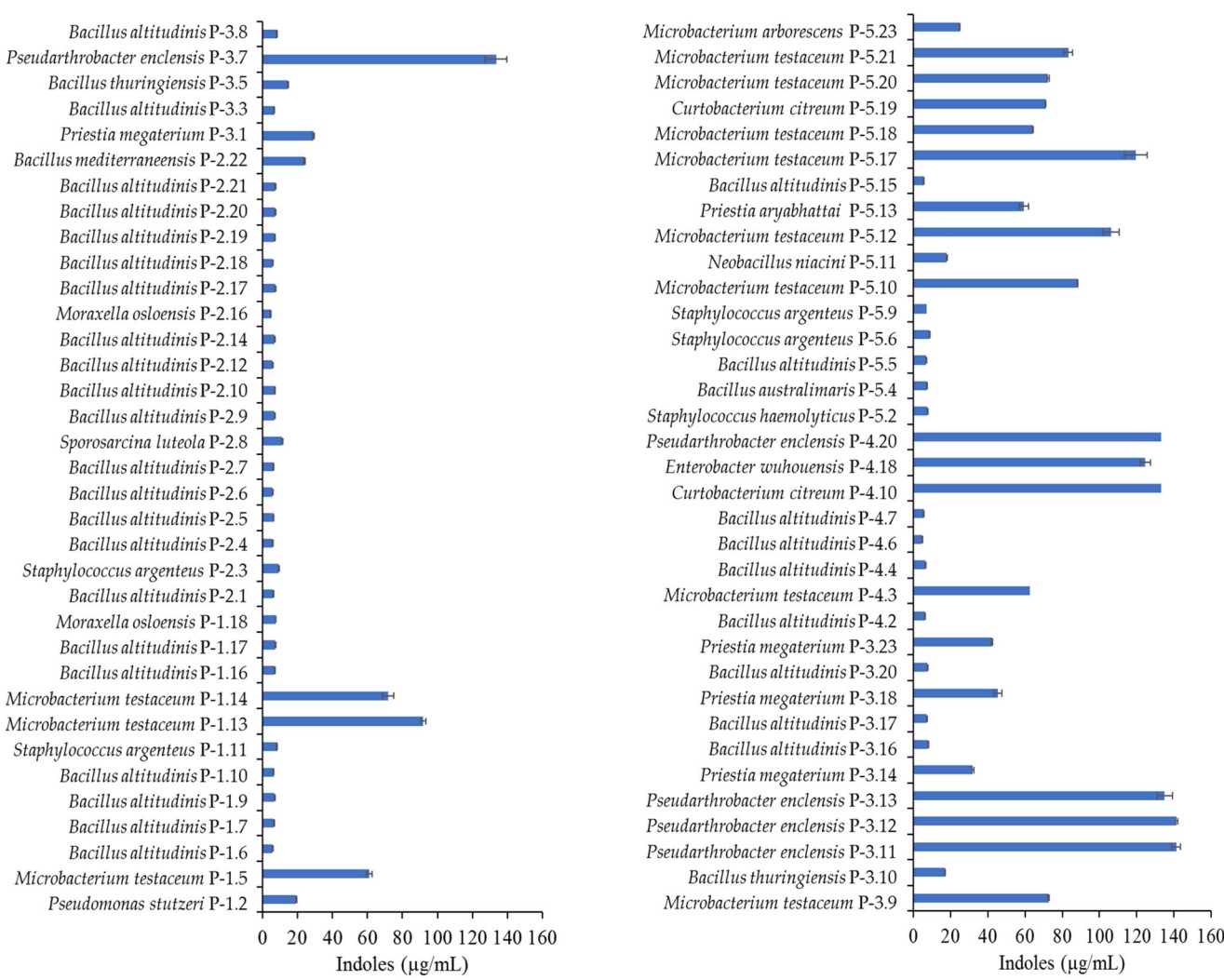

**Figure 3.** Indole-related compounds production and accumulation of 70 bacterial endophytes from tea leaves cultivated in nutrient broth supplemented with 1 mg/mL L-tryptophan at 30 °C at 150 rpm for 72 h.

### 3.3. Investigation of Inorganic Phosphate Solubilization and Siderophore Production

Out of 70 bacterial isolates, the majority of bacterial endophytes recovered from tea leaves (55 isolates or 78.57%) were positive for inorganic phosphate solubilization as they produced clear zones surrounding their colonies on phosphate-supplemented medium (Figure 4A). Moreover, *Bacillus altitudinis* P-4.7, *Enterobacter wohouensis* P-4.18, and *Staphylococcus haemolyticus* P-5.2 showed a large diameter of clear zone up to 4 mm. The appreciable phosphate solubilization activity was also displayed by *Staphylococcus argenteus* P-2.3, *Bacillus safensis* P-5.4, and *Curtobacterium citreum* P-5.19. Significant production of siderophore by endophytic bacterial isolates was observed in 64 of the 70 bacterial isolates, as clearly evidenced by a visible halo around colonies in the CAS medium. Among bacterial endophytes capable of siderophore production, *Bacillus altitudinis* P-2.12, *Priestia megaterium* P-3.14, *Microbacterium testacum* P-4.3, and *Neobacillus niacin* P-5.11 showed notable siderophore production.

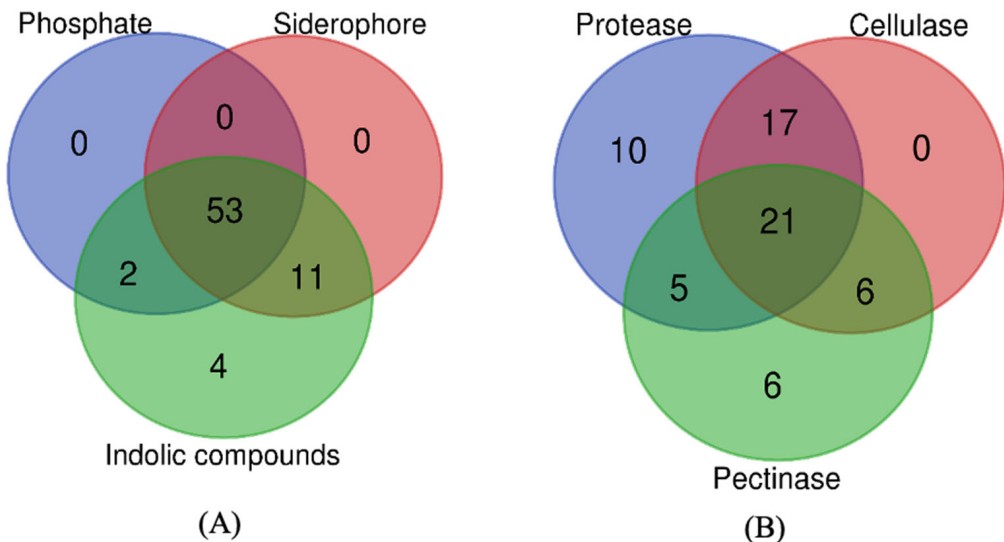

**Figure 4.** Venn diagram representation of the plant-growth-promoting traits (**A**) and enzyme activity (**B**) of bacterial endophytes from tea leaves.

*3.4. Extracellular Hydrolytic Enzyme Production and Tannin-Tolerant Characteristics*

All 70 endophytic bacterial isolates were investigated for their ability to produce extracellular hydrolytic enzymes such as cellulases, proteases, and pectinases, and the results are presented as a Venn diagram (Figure 4B). A total of 54 isolates (77.14%) showed protease production capacity, 46 isolates (65.71%) showed the ability to produce cellulases, while 37 isolates (52.85%) were positive for pectinase production. However, there was no significant correlative relationship between the production ability of these three hydrolytic enzymes among all recovered bacterial endophytes. The tannin tolerance activity of isolated bacterial endophytes was of interest as 80% (56 from 70 isolates) of all bacterial endophytes were capable of growth on nutrient agar supplemented with 10–50 g/L tannic acid (Figure 5). Remarkably, *Moraxella osloensis* P-1.18, *Bacillus thuringiensis* P-3.5, *Pseudarthrobacter enclensis* P-3.12, *Enterobacter wohouensis* P-4.18, and *Microbacterium testacum* P-5.17 were highly tolerant up to 50 g/L tannic acid.

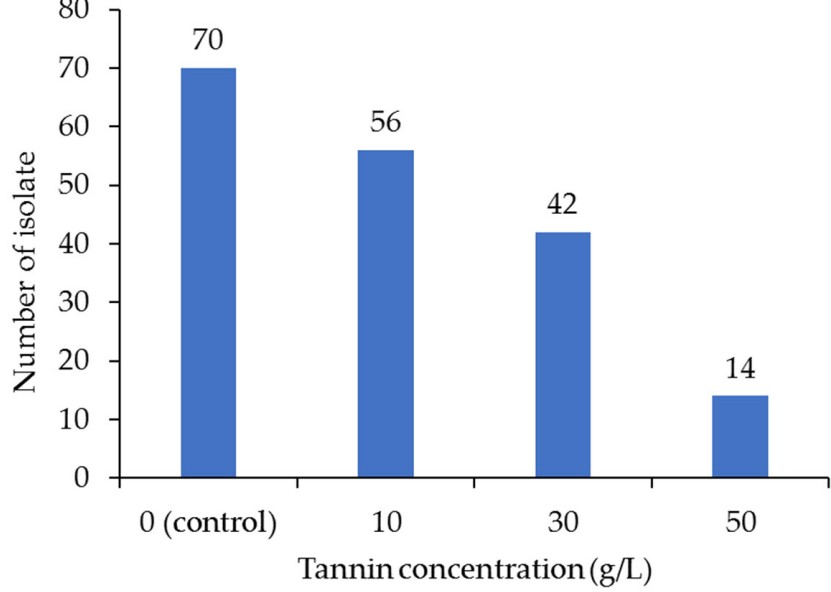

**Figure 5.** Number of bacterial endophytes from tea leaves with tannin tolerance capacity after cultivation on nutrient agar supplemented with 10, 30, and 50 g/L tannic acid at 30 °C for 72 h.

### 3.5. Confirmation of IAA Production from HPLC Analysis

Based on indole-acetic acid (IAA) estimation using HPLC analysis, a total of seven isolates that produced and accumulated the highest concentrations of indoles were confirmed, namely *Pseudarthrobacter enclensis* P-3.7, *Pseudarthrobacter enclensis* P-3.11, *Pseudarthrobacter enclensis* P-3.12, *Pseudarthrobacter enclensis* P-4.20, *Enterobacter wuhouensis* P-4.18, *Curtobacterium citreum* P-5.19, and *Microbacterium testaceum* P-5.12. The results showed significant difference in IAA production as presented in Table 2. *C. citreum* P-5.19 produced and accumulated IAA at the highest concentration up to 367.59 (µg/mL) followed by *P. enclensis* P-3.12, *P. enclensis* P-3.7, *P. enclensis* P4.20, *M. testaceum* P-5.12, *E. wuhouensis* P-4.18, and *P. enclensis* P-3.11. Thus, *C. citreum* P-5.19 and *P. enclensis* P-3.12 were selected for subsequent experiments to determine the effects of IAA produced by these bacteria on seed germination.

**Table 2.** Indole-3-acetic acid (IAA) levels in culture supernatant of selected endophytic bacterial isolates determined by HPLC analysis.

| Bacterial Isolates | Amount of IAA (µg/mL) |
|---|---|
| *Pseudarthrobacter enclensis* P-3.7 | 215.99 ± 3.23 |
| *Pseudarthrobacter enclensis* P-3.11 | 122.81 ± 1.67 |
| *Pseudarthrobacter enclensis* P-3.12 | 266.97 ± 2.09 |
| *Enterobacter wuhouensis* P-4.18 | 148.91 ± 1.17 |
| *Pseudarthrobacter enclensis* P-4.20 | 170.38 ± 1.12 |
| *Curtobacterium citreum* P-5.19 | 367.59 ± 2.45 |
| *Microbacterium testaceum* P-5.12 | 171.72 ± 2.31 |

### 3.6. In Vitro Plant-Growth-Promoting Test

Two bacterial isolates *C. citreum* P-5.19 and *P. enclensis* P-3.12 were selected for the plant-growth-promoting characteristics on tomato and sunflower seeds in a Petri dish experiment (Figure 6). Seed inoculation with the selected isolates showed significantly ($p < 0.05$) better germination rate, shoot height, root length, fresh weight, and dry weight of seedlings compared with the untreated control groups (Table 3). Additionally, the vigor index was significantly higher (>2-fold) in both treatment groups compared to controls for the sunflower and tomato experimental units, respectively.

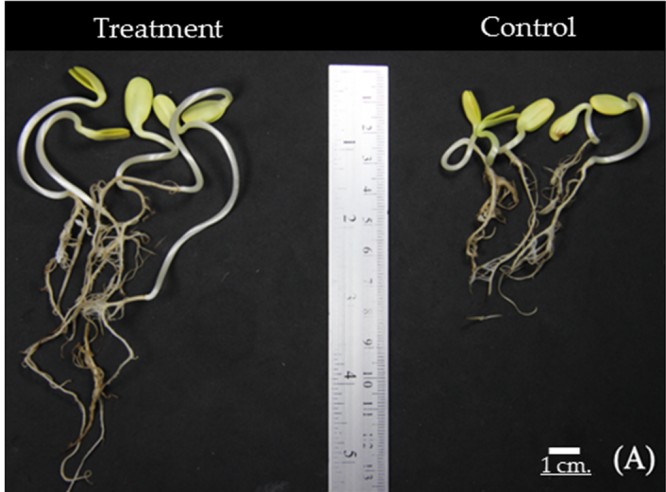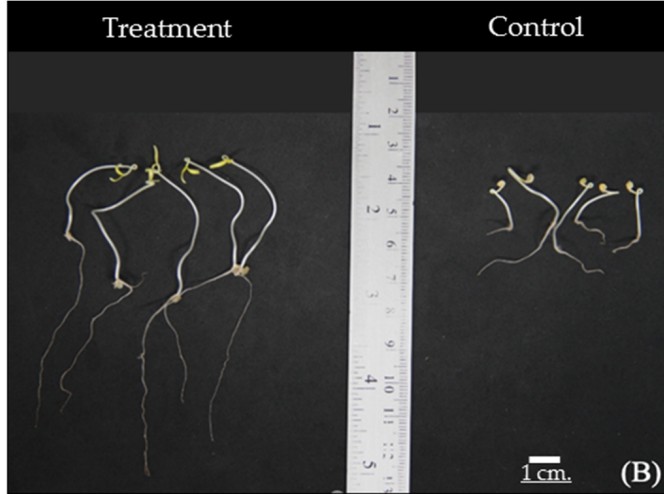

**Figure 6.** Effect of seed treatment with viable cells of selected bacterial endophyte (*Curtobacterium citreum* P-5.19) on shoot and root growth of sunflower (**A**) and tomato (**B**) seedlings. Photographs showing higher shoot and root growth in bacteria-treated seedlings compared to untreated seedlings. Photographs were taken at day 5 after the treated seeds were placed in Petri dishes for germination.

**Table 3.** Effects of *Pseudarthrobacter enclensis* P-3.12 and *Curtobacterium citreum* P-5.19 viable cells on sunflower and tomato seed germination and growth parameters under in vitro axenic conditions.

| Parameters | Sunflower Seeds | | | Tomato Seeds | | |
|---|---|---|---|---|---|---|
| | Control | Treatment | | Control | Treatment | |
| | | P-3.12 | P-5.19 | | P-3.12 | P-5.19 |
| Germination (%) | 65.50 ± 10.54 [b] | 98.33 ± 0.47 [a] | 97.67 ± 4.30 [a] | 45.00 ± 4.99 [c] | 53.33 ± 3.29 [b] | 92.00 ± 6.64 [a] |
| Vigor index | 250.86 ± 54.93 [c] | 756.16 ± 19.76 [a] | 587.98 ± 29.49 [b] | 239.05 ± 72.81 [c] | 454.37 ± 51.26 [b] | 829.84 ± 54.49 [a] |
| Root length (cm) | 2.13 ± 0.46 [c] | 4.89 ± 1.01 [a] | 3.31 ± 0.75 [b] | 2.47 ± 0.73 [b] | 4.48 ± 0.27 [a] | 4.29 ± 0.24 [a] |
| Shoot length (cm) | 1.70 ± 0.29 [b] | 2.80 ± 0.54 [a] | 2.71 ± 0.47 [a] | 2.85 ± 0.35 [c] | 4.04 ± 0.25 [b] | 4.73 ± 0.33 [a] |
| Fresh weight (mg) | 167.40 ± 0.21 [c] | 223.83 ± 0.02 [b] | 283.8 ± 0.31 [a] | 36.80 ± 0.03 [c] | 40.93 ± 0.03 [b] | 61.80 ± 0.07 [a] |
| Dry weight (mg) | 53.20 ± 0.06 [c] | 59.9 ± 0.08 [b] | 69.9 ± 0.05 [a] | 4.00 ± 0.00 [b] | 5.40 ± 0.21 [b] | 7.60 ± 0.24 [a] |

Note: Data are presented as mean ± SD of three replications. Superscripted letters refer to significantly different mean values ($p < 0.05$) of each parameter.

PC 1 and PC 2 in Figure 7 explained a cumulative ~81.28% variance in the data for sunflower seeds treated with *C. citreum* P-5.19, *P. enclensis* P-3.12, and the uninoculated control group. Similarly, ~89.58% of cumulative variance was explained for the tomato seed experiment. This implies that more than 80% of the information was captured by the statistical algorithm, that explained the significance of treatment in sunflower and tomato seed germination experiments, which is sufficient for descriptive purposes. In Figure 7A,B, *C. citreum* P-5.19 and *P. enclensis* P-3.12 clustered separately from the uninoculated control groups, which correlates with the significant differences observed in one-way ANOVA for treatment groups compared to the control groups (Table 3).

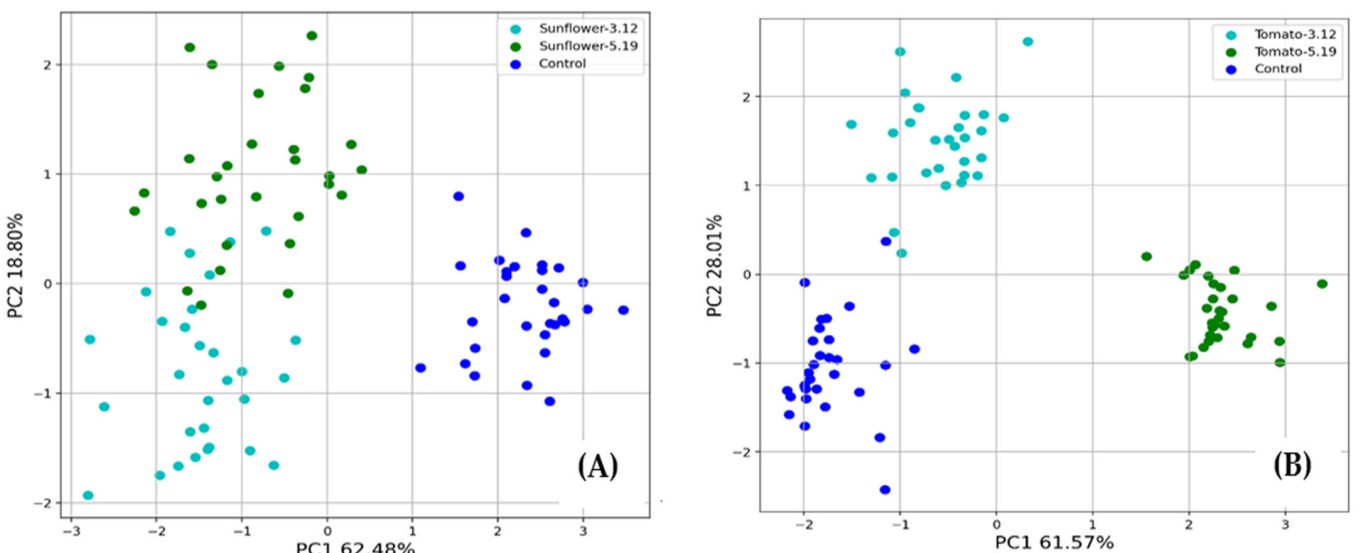

**Figure 7.** Clustering relationship of inoculated sunflower seeds (**A**) and tomato seeds (**B**) with isolates of *Pseudarthrobacter enclensis* (P-3.12) and *Curtobacterium citreum* (P-5.19) and co-inoculation with uninoculated control according to the PCA analysis of the growth-promoting experiment.

## 4. Discussion

The inherent potential shown by endophytic bacteria has prompted a need to invest research efforts into further understanding their microbial diversity as well as broadening their scope of application, which includes countering insect and plant pathogens as well as managing environmental stresses [33]. Consequently, the importance of finding new and favorable endophytic microorganisms from a diverse range of plants in various ecosystems is a promising pursuit. Endophytic bacteria have traditionally been investigated using culture-dependent techniques, even though only ~1% of bacteria are culturable [34]. In the present study, we isolated endophytic bacteria from tannin-rich leaves of tea plants

(*Camellia sinensis* var. *assamica*) considering the dearth of information regarding endophytic bacterial community inhabiting tea leaves as it relates to their functional attributes and application in sustainable agriculture. This study clearly showed the predominant existence and wide abundance of the genus *Bacillus* among all the tea gardens. Among the *Bacillus* genus, *Bacillus altitudinis* species was determined to be the dominant and frequently found in Miang leaves collected from Miang tea garden in Pa Pae sub-district, Mae Taeng district, Chiang Mai, which is reported to be the source of high-quality young tea leaves, a raw material for Miang production through non-filamentous fungi growth-based fermentation (NFP) in Northern Thailand [18]. *Bacillus altitudinis* has been previously reported as an endophytic bacterium in indigenous black rice in India [35] and showed bioprotectant activity against plant pathogenic fungi [36]. Previous studies have reported similar results where *Bacillus* spp. was indicated as a dominant endophytic bacteria in plants such as ginseng [37], banana [38], and mulberry [39]. In addition, it was discovered that *Bacillus* species are common colonizers of various plants due to their strong environmental adaptability [40].

Among 70 endophytic bacterial isolates reported in this study, some isolates were identified to be the same species. However, considering that the five tea gardens from which the tea leaves were collected were separated by distances averaging 2–5 km, in addition to the varying plant growth promotion properties observed among these closely related (same species) endophytes as shown in Table S1, we assumed that although these confounding isolates were identified to be the same species, the variations observed in the supplementary table are enough to confirm their differences. Nevertheless, from a biodiversity viewpoint, further identification of the differences among the same species, especially *Bacillus altitudinis*, is needed to ascertain the grouping at species and strain levels using special techniques such as random amplified polymorphic DNA (RAPD) profiling. Moreover, the use of other gene sequences such as RNA polymerase sigma factor gene (*rpoD*) and DNA gyrase gene (*gyrA*) would be helpful for the better resolution of bacterial strains at the species level. The results after grouping may provide a better overall image of tea endophytic bacterial diversity, which is also the next target of our research group.

Indole-acetic acid (IAA) is one of the most important auxins synthesized by a variety of endophytic bacteria. This phytohormone affects plant cell division, differentiation, and cell elongation; stimulates seed germination; and also affects biosynthesis of metabolites, pigment formation, and photosynthesis [41]. Endophytic bacteria discovered in this study produced indole compounds ranging from 4.65 to 141.47 μg/mL using tryptophan as a precursor, which is similar to bacterial isolates from aloe vera [42] and ryegrass [43]. Among all these isolates, high production of indole compounds was observed in the *Pseudarthrobacter* genus, which appeared to produce the highest levels of indoles in this study, although several reports have implicated *Microbacterium* isolates from tea with significant production of IAA [44,45], while *Enterobacter* was the predominant IAA producer among aloe vera isolates [42], which is in line with the findings of this study. The phosphate-solubilizing ability of bacterial endophytes has gained much interest in agriculture, as it can increase the availability of mineral phosphate to the plant [46]. Our findings demonstrated that the major phosphate-solubilizing bacteria belonged to the genus *Bacillus*. *Bacillus altitudinies*, *Staphylococcus haemolyticus*, and *Enterobacter wuhouensis* had higher phosphate solubilization activity. *Microbacterium testecum*, *Staphylococcus aureus*, and *Cartobacterium citrium* also showed appreciable ability to solubilize inorganic phosphate. Earlier studies reported that bacteria belonging to genera *Bacillus*, *Enterobacter*, *Microbacterium*, and *Cartobacterium* can solubilize insoluble phosphate, which agrees with our findings [42,43,47]. However, further experiments are needed for determining safety of application of some bacterial isolates previously reported as pathogenic bacteria, such as *S. haemolyticus* and *S. aureus*. Bacterial siderophores can promote plant growth directly by increasing iron availability to plants and indirectly depriving fungal pathogens of this essential ion [48]. Endophytic bacterial isolates from this study also produced promising siderophores, implying that isolates were also capable of inhibiting deleterious microorganisms. Yu et al. [49] reported *Bacillus* spp. as effective siderophore producers, which agrees with our study. When considering the

three important characteristics described above, there are interesting correlative relationships between the capability of inorganic phosphate solubilization, indole compounds, and siderophore production as 53 from 70 or 75.71% of endophytic bacterial isolates are positive for all three-plant-growth-promoting activities (Figure 4). This confirms the positive potential of bacterial endophytes discovered from this study.

Endophytic microbes live in plant tissues without causing substantive harm to the host. They exist within the living tissues of most plant species in the form of symbiotic associations to slightly pathogenic; additionally, endophytes may modify their genome in order to adapt to or avoid the host defense mechanisms [33]. Production of hydrolytic enzymes such as protease, cellulase, and pectinase is recognized as the prominent functional traits for indirect plant growth promotion, and these enzymes are essential during the colonization and migration of endophytes through the degradation of cell walls [12,50]. Although beneficial bacteria spread and enter the inner plant tissue using the same entry mechanisms as the bacterial pathogens, the host develops strategies to allow particular bacterial genera.

The majority of our bacterial isolates were capable of secreting lytic enzymes as has been corroborated from previous findings, where bacterial endophytes *Bacillus* and *Microbacterium* showed the ability to synthesize hydrolytic enzymes [42,51]. Tannin is the most abundant phenolic compound found in tea leaves and is known to be a growth inhibitor of certain microorganisms [52]. The ability to tolerate tannin is assumed to be one of the most essential factors for developing a bacterial community in tea plants. In our current study, most of the endophytic bacterial isolates had tannin tolerance ability, which also indicated that the tea leaves can be a selective microhabitat for bacteria that can survive in tannin-rich substrate. This was supported by a study conducted by Rungsirivanich et al. [53] as they discovered that *Bacillus* spp. isolated from tea leaves could be cultured on TSA supplemented with tannic acid.

Plant growth promotion and protection from diseases achieved via plant interaction with beneficial bacterial endophytes is a well-established phenomenon in modern agricultural practice [54]. Furthermore, production of phytohormones also accelerates the action of specific enzymes (e.g., amylase), which activate early germination and increase the availability of starch assimilation [55]. Plants inoculated with IAA-producing PGP bacteria alter root architecture by accelerating root hair formation and increasing the number and length of primary and lateral roots, thereby enhancing the root surface area for mineral uptake and exudation [41,56]. Based on the important characteristics of plant-growth-promoting bacteria, especially the capability of IAA production confirmed by HPLC analysis, the bacterial endophytes *C. citreum* P-5.19 and *P. enclensis* P-3.12 were selected to investigate the effects of these endophytic bacteria on plant-growth-promoting activity via the microbial treatments on tomato and sunflower seed germination where the treatment groups significantly ($p < 0.05$) increased germination rate and enhanced vigor and growth of seedlings compared to the untreated control, which has been similarly observed for tomato seeds inoculated with *Bacillus* sp. [32] and *Azotobacter* sp.-inoculated sunflower seeds [57]. Although the mechanism of growth enhancement was beyond the scope of our investigation, it suffices to suggest that phytohormones such as IAA produced by these isolates significantly contributed to plant growth promotion. Furthermore, the improved growth parameters observed for sunflower and tomato treated groups as opposed to the uninoculated controls (Table 2) were also corroborated by the multivariate principal component analysis (PCA) algorithm as both treatments clustered separately from the controls, even more so for tomato seeds, which suggests the positive effects of bacterial association with the seeds (Figure 7). Figure 7A shows a light intersection between *P. enclensis* P-3.12 and *C. citreum* P-5.19-treated sunflower seeds, which suggests that *P. enclensis* P-3.12 and *C. citreum* P-5.19 may mediate plant growth promotion through separate mechanisms that bridge at certain metabolic intermediaries. This may not be the case for tomato-treated seeds as the clusters are quite distinct. Overall, this bacterial isolate has demonstrated notable plant-growth-promoting abilities and could be further explored to understand

the molecular mechanisms of action as well as means to improve its applications both in agriculture and wider industries for improving plant growth and production.

## 5. Conclusions

This present study demonstrates the presence of culturable endophytic bacteria in tea leaves collected from Northern Thailand. *Bacillus* was found as the predominant genus out of 11 endophytic bacterial genera, with *Bacillus altitudinis* constituting the largest identified species. The highest IAA production was observed in *Curtobacterium citreum* P-5.19, followed by *Pseudarthrobacter enclensis* P-3.12, *P. enclensis* P-3.12, and *C. citreum* P-5.19, which also showed the potential to be used as bio-formulated fertilizer for sustainable crop production. The findings of the interesting endophytic bacteria from tea plants collected from the selected tea plantation site support the hypothesis of the involvement of endophytic microbes in a traditional tea garden in Northern Thailand without the use of agrochemicals, which is an efficient agroforestry model for a sustainable agricultural system. However, further investigations as well as the molecular mechanisms to assess the involvement of the potent endophytic bacteria in the tea plant host are required before recommending or applying these selected microbes in biofertilizer formulations.

**Supplementary Materials:** The following are available online at https://www.mdpi.com/article/10.3390/agriculture13030533/s1, **Table S1**. Characteristics based on plant-growth-promoting properties, hydrolytic enzyme production and tannin tolerances of endophytic bacteria isolated from tea leaves.

**Author Contributions:** Conceptualization, M.H.K. and C.K.; methodology and formal analysis, M.H.K., K.U., P.K., P.W., S.L. and R.K.G.; investigation, M.H.K., K.U., N.S. and C.K.; writing—original draft preparation, M.H.K., K.U. and C.K.; writing—review and editing, M.H.K., S.L., K.S. and C.K.; supervision, C.K. All authors have read and agreed to the published version of the manuscript.

**Funding:** This work was partially supported by the Office of Research Administration, Chiang Mai University.

**Institutional Review Board Statement:** Not applicable.

**Data Availability Statement:** Not applicable.

**Acknowledgments:** The authors would like to acknowledge Chiang Mai University's Presidential scholarship program. We also acknowledge Faculty of Agro-Industry, Chiang Mai University, for research facilities.

**Conflicts of Interest:** The authors declare no conflict of interest.

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
