# Peer review of "Endophytic Bacteria Isolated from Tea Leaves (Camellia sinensis var. assamica) Enhanced Plant-Growth-Promoting Activity"

_agriculture, doi:10.3390/agriculture13030533_

Round 1

Reviewer 1 Report

This manuscript describes the isolation and characterization of endophytic bacteria from Tea leves in Thailand. The authors found  number of bacteria that have favorable features as biofertilizers. The content is novel and worth publication. Hoever, the manuscript should be revised before the publication, because of many errors, such as mis spelling, format and redundancy.      L64 Enterobacter shoud be in Italic.     L77 "with direct financial imperatives of tea products" Are there any significant relationship with financial imperatives to this study?     L131 delete "and genus"     Fig. 1: divided into two parts? I guess it should be connected.   L284 "Pseudarthrobacter sp. (P-4 and P-5) and Moraxella sp. (P-1 and P-2) were also found in other tea gardens (Figures 2A and 2B)." I can not understand this sentence. Pseudarthrobacter was founfd only in P4 and P5. Moreover, Moraxella did not appear in the figure.     L307 "Investigation of inorganic phosphate solubilization and siderophore production" should be in italic.     L316 Delete "There were  Only 6 isolates were negative in siderophore production." Even kids can do such calculation.     Figure 5. The title of X axis should be revised. I think it is not showing the Tannin concentration, but the maximum tannnin concentration of a bacteria can be tolerant.     L350 insert "production" after "IAA".     Fig. 6. Does the photo showing the seeds treated with both bacteria species or only one bacterial species?     L382-390 and Fig. 7 The significance of thegrowth promoting effect is clear enough in Table 2, and these parts are not essential.     Discussion part is too long because it containig redundant expression of other parts. It shoud be thoroughly revised. For example, L404-412, the sentences describing about the NGS should be deleted, because there is no results or comparison with NGS has been done in this paper. In addition, L418-421, "and the most frequently identified genus was Bacillus ----- and Sporosarcina (Figure 2) " is redundant and not nesessary.      L434 "The existence of bacterial endophytes in the Bacillus and Pseudomonas genera make them amenable to easy culturing and thus can be isolated via less expensive culture dependent techniques [43]. ": Does this sentence mean that the frequency of isolation of Bacillus was affected by its culturability? Then please make it clear. Otherwise, this sentence is not meaningful and should be deleted.     L443-445 "All endophytic bacterial isolates discovered from this study were able to produce significant amounts of in dole compounds (IAA, IBA, and other derivatives) with tryptophan as a precursor where the indoles ranged from 4.65 to 141.47 μg/mL. ": Revise the explanation. Not all isolates produced siginificant amout of Indole compounds.     L455-466: What do you want to say by this paragraph? What is the conclusion? Difference in the concentration was already clear in the Results part.     L535-547: not required, and should be deleted.     L550-567: Make it more concise by deleting the redundant expression.

Reviewer 2 Report

§  The concept of exploiting the endophytic microbe for improving the growth and quality of plants is good one.  However, the work cannot be accepted in its current state. MS should revise before being submitted again for review.

Comment

§  Mentioned longitude and latitude and altitude positions along with average rainfall, weather, and temperature of places from where the samples were collected for isolation of endophytes.

§  MS have too many long and repeated phrases. This need to be revised.

§  Include the company names and the nation from where the equipments were purchased. Whenever applicable.

§  Did the authors submit NCBI the 16S sequences? If so, please include the accession number in the new table of BLAST results that includes the name and percent identity with most similar organisms. If the authors continue to withhold the 16S sequences, they will first submit them to NCBI to obtain an accession number and construct a table of the BLAST results. To ensure differentiation and avoid the replication of identical isolates, this is crucial.

§  Clastal W is an integrated programme in MEGA software. The authors employed the bioedit programme for what reason? Further Actinobacteria are positioned between the firmicutes groups in the phylogenetic tree. Phylogenetic tree does not appear to be correct as a result. Please double-check and re-create.

§  Did the author trim the flanking sequences after multiple sequence alignment? How many aligned nucleotide were taken for preparation of phylogenetic tree.

§  The boot-strap probability for construction of phylogenetic tree should be mentioned in caption of phylogenetic figure.

§  The meaning of scale bar should be mentioned in caption of phylogenetic figure

§  Authors mentioned that endophytes exist within host plant tissue. The authors were isolated the endophytes that have potential to produced the cellulase, an enzyme that are responsible to degrade the cellulose, a main constituent of plant cell wall. Kindly explain this in discussion section that how the plant cell survive from such type of hydrolytic enzymes of endophytes especially from cellulase.

§  References somewhere are not correct format. Kindly check it.

Reviewer 3 Report

As a reviewer, I carefully studied the article entitled" Endophytic Bacteria Isolated from Tea Leave (Camellia sinensis var. assamica) Enhanced Plant Growth Promoting Activity" and came to the following conclusion.

the study has no novelty and is just routine work, however, it can be considered if improved according to the suggestions.

(1) the appropriate objective is missing, and the study seems like collection of random premature experiments. 

(2) the authors should formulate a proper story with clear objectives and substantial experimental work.

(3) the molecular machinery needed to be asses in bacteria and the plant and proper crosstalk of the different molecular mechanisms should be mentioned.

Round 2

Reviewer 1 Report

The paper is correctly revised and can be published in the journal.

Author Response

Thank you for your kind consideration to accept our manuscript for publication.

Reviewer 2 Report

The BLAST result clear cut show the drawback of the MS.  The authors made wrong claim to number of endophytes isolated . The number of endophytes claimed to be isolated by authors is not justify by the BLAST table. The table clear cut shows that most of the isolated strain are actually replicated strains instead of isolated strain.

such as strain P1.6, P1.7, P1.9, P1.10, P1.16, P1.17, P2.1, P2.4, P2.5, P2.6, P2.7, P2.9, P2.10, P2.12, P2.14, P2.17, P2.18, P2.19, P.2.20, P2.21, P3.3., P3.8, P3.16, P3.17, P3.20, P4.2, P4.4, P4.6, P4.7, P5.5, P5.15 are the replicated strain of same endophytes and match with single microbial strain Bacillus altitudinis 41KF2b in BLAST result.

Similarly, P-1.5, P3.9, P4.3, P5.10, P5.12, P5.17, P5.18, P5.20, P5.21 are the replicated strain of same endophytes and match with single microbial strain Microbacterium testaceum NBRC 12675. 

P4.10 and P5.19 are the replicated strain of same endophytes and match with single microbial strain Curtobacterium citreum DSM 20528.

P1.11, P2.3, P5.6, P5.9 are the replicated strain of same endophytes and match with single microbial strain Staphylococcus argenteus MSHR1132.

P3.1, P3.14, P3.18, P3.23 are the replicated strain of same endophytes and match with single microbial strain Priestia megaterium NBRC 15308

P3.7, P3.11, P3.12, P3.13, P4.20  are the replicated strain of same endophytes and match with single microbial strain Pseudarthrobacter enclensis NIO-1008

P1.3, P1.14 are the replicated strain of same endophytes and match with single microbial strain Microbacterium testaceum BJML01000022.

P1.11, P2.3, P5.6, P5.9 are the replicated strain of same endophytes and match with single microbial strain Staphylococcus argenteus MSHR1132

P1.18 and P2.16 are the replicated strain of same endophytes and match with single microbial strain Moraxella osloensis CCUG 350.

P3.5 and P3.10 are the replicated strain of same endophytes and match with single microbial strain Bacillus thuringiensis ATCC10792

So in my think this is fabricated MS and show false data

Reviewer 3 Report

The authors successfully added the suggestions and comments and therefore, I accept the manuscript. after improving the following

1. The figure resolution must be improved.

Author Response

1. The original files of all figures have been improved.

2. Thank you for your kind consideration to accept our manuscript for publication.